# Competing Endogenous RNA Networks as Biomarkers in Neurodegenerative Diseases

**DOI:** 10.3390/ijms21249582

**Published:** 2020-12-16

**Authors:** Leticia Moreno-García, Tresa López-Royo, Ana Cristina Calvo, Janne Markus Toivonen, Miriam de la Torre, Laura Moreno-Martínez, Nora Molina, Paula Aparicio, Pilar Zaragoza, Raquel Manzano, Rosario Osta

**Affiliations:** 1Department of Anatomy, Embryology and Animal Genetics, University of Zaragoza, Centro de Investigación Biomédica en Red de Enfermedades Neurodegenerativas (CIBERNED), Agroalimentary Institute of Aragon (IA2), Institute of Health Research of Aragon (IIS), Calle Miguel Servet 13, 50013 Zaragoza, Spain; leticiamoreno@unizar.es (L.M.-G.); tlopez@unizar.es (T.L.-R.); accalvo@unizar.es (A.C.C.); toivonen@unizar.es (J.M.T.); mtorre@unizar.es (M.d.l.T.); lauramm@unizar.es (L.M.-M.); noramolinat@gmail.com (N.M.); paula_bureta@hotmail.com (P.A.); pilarzar@unizar.es (P.Z.); rmanzano@unizar.es (R.M.); 2Geriatrics Service, Hospital Nuestra Señora de Gracia, Calle Ramón y Cajal 60, 50004 Zaragoza, Spain

**Keywords:** competing endogenous RNAs (ceRNA), neurodegenerative diseases (NDDs), extracellular/circulating biomarkers, microRNA, long non-coding RNA, circular RNA, pseudogene, mRNA, ceRNA network (ceRNET), RNA editing

## Abstract

Protein aggregation is classically considered the main cause of neuronal death in neurodegenerative diseases (NDDs). However, increasing evidence suggests that alteration of RNA metabolism is a key factor in the etiopathogenesis of these complex disorders. Non-coding RNAs are the major contributor to the human transcriptome and are particularly abundant in the central nervous system, where they have been proposed to be involved in the onset and development of NDDs. Interestingly, some ncRNAs (such as lncRNAs, circRNAs and pseudogenes) share a common functionality in their ability to regulate gene expression by modulating miRNAs in a phenomenon known as the competing endogenous RNA mechanism. Moreover, ncRNAs are found in body fluids where their presence and concentration could serve as potential non-invasive biomarkers of NDDs. In this review, we summarize the ceRNA networks described in Alzheimer’s disease, Parkinson’s disease, multiple sclerosis, amyotrophic lateral sclerosis and spinocerebellar ataxia type 7, and discuss their potential as biomarkers of these NDDs. Although numerous studies have been carried out, further research is needed to validate these complex interactions between RNAs and the alterations in RNA editing that could provide specific ceRNET profiles for neurodegenerative disorders, paving the way to a better understanding of these diseases.

## 1. Introduction

Neurodegenerative diseases (NDDs) are of increasing relevance in public health due to the aging of the global population. These common and complex disorders are characterized by a progressive and selective loss of neurons from specific regions of the central nervous system (CNS), the most prevalent NDDs being Alzheimer’s disease (AD), Parkinson’s disease (PD) and amyotrophic lateral sclerosis (ALS). Although protein aggregation is a common hallmark for these disorders, there is growing evidence that alterations in RNA metabolism contribute to the etiopathogenesis of NDDs [1,2,3]. Defects at all levels of gene regulation, from RNA synthesis to degradation, have been associated with disease-specific alterations in RNA-binding proteins (RBPs) and non-coding RNAs (ncRNAs) [2,3]. Interestingly, approximately 80% of the human genome is transcribed as non-coding transcripts, whereas only 2% encodes proteins [4], highlighting the potential of ncRNAs as disease modifiers, and, given their particular abundance in the CNS, their potential contribution to NDDs onset and development [5,6]. 

ncRNAs can be classified into two groups according to their length: small ncRNAs (<200 nucleotides) and long ncRNAs (>200 nucleotides) [7]. Among small ncRNAs, microRNAs (miRNA) stand out, being around 22 nucleotides long and regulating gene expression at the post-transcriptional level in a sequence-specific manner [8]. Approximately 70% of the identified miRNAs are expressed in the brain [9] and have been described as major regulators of neuronal homeostasis, their misregulation being associated with pathological conditions of CNS [8]. The largest class of ncRNAs in the mammalian genome is long ncRNAs (lncRNAs), which can be further grouped into linear RNAs and circular RNAs [7,10]. Linear lncRNAs (hereon referred to as lncRNAs) are similar to protein-coding messenger RNA (mRNA) in sequence length and transcriptional and post-transcriptional behavior [7]. However, lncRNAs play a different cellular role compared to mRNAs. Moreover, they have been described to be involved in brain development, neuronal function, maintenance and differentiation [5]. Circular RNAs (circRNAs) represent a relatively recently discovered class of RNAs that, unlike linear RNAs, are characterized by a covalent bond that joins the 5′ and 3′ ends and confers increased stability (half-life of 48 h vs. 10 h for mRNAs) [11]. circRNAs are highly abundant in the brain, enriched in synaptoneurosomes and upregulated during neuronal differentiation [12], so they could be promising biomarkers in age-associated NDDs.

On the other hand, a considerable number of pseudogenes can be transcribed to ncRNAs, even though they have historically been regarded as inactive gene sequences [13,14]. In fact, there is mounting evidence that pseudogenes may modulate the expression of parental as well as unrelated genes [13,14]. Therefore, alteration of pseudogene transcription could perturb gene expression homeostasis leading to disease [13]. 

In 2011, Pier Paolo Pandolfi’s group proposed the so-called ceRNA hypothesis [15], which sought to explain how RNAs “talk” to each other, establishing interactions that modify functional genetic information and that may play major roles in pathological conditions. This hypothesis is based on the fact that miRNAs can recognize their specific target sites called miRNA response elements (MRE) in different RNA molecules, causing target repression via miRNA-RISC complex-mediated degradation. Thereby, miRNAs could mediate regulatory crosstalk between the diverse components of the transcriptome, comprising mRNAs and ncRNAs, which include pseudogenes, lncRNAs and circRNAs.

In a simplified manner, when two RNA molecules share the same MRE they potentially compete for the same pool of miRNAs. Thus, when the expression of a ceRNA is upregulated, it will bind and titrate more miRNAs (phenomenon called miRNA sponging), leaving fewer miRNA molecules available for binding the mRNA with shared MRE. Hence, this corresponding mRNA will become derepressed. In reverse, when the ceRNA levels are reduced as a consequence of a biological disturbance, the corresponding mRNA will be downregulated due to hyperrepression (Figure 1). 

Without doubt, the reality is more complex and a miRNA can bind more than one mRNA (50% of miRNAs are predicted to target 1–400 mRNAs and some of them up to 1000) [16]. Likewise, most ceRNAs contain 1 to 10 MREs [16] and, as a consequence, complex ceRNA networks involving a large number of RNA molecules are established. Novel bioinformatic and computational tools have enabled to elucidate an increasing number of ceRNA networks, as well as predict the most important enclaves of them. These may provide a valuable global vision to identify new biomarkers, underlying pathways or potential therapeutic targets for complex disorders such as NDDs.

The potential of ceRNAs as biomarkers, the major focus of this review, is augmented by the fact that ncRNAs have been found in body fluids like blood or urine free or inside extracellular vesicles including exosomes, which would allow obtaining new biomarkers in a non-invasive way. On this basis, emerging research on the role of ncRNAs in various diseases has arisen [17,18,19]. In this review, the potential of ceRNA networks as biomarkers in neurodegenerative diseases is discussed.

## 2. ceRNA Networks and Neurodegenerative Diseases

Over the last years, the ceRNA hypothesis has been corroborated by a large number of experiments. However, investigation of ceRNA mechanisms and their interaction networks has been mainly carried out in cancer research [20,21,22,23]. Nevertheless, some advances have also been made in the field of NDDs. Here, we aim to review the ceRNA networks (ceRNETs) reported to date in experimental (Table 1), and transcriptome profiling and bioinformatic studies (Table 2) in this field, classifying them according to the NDD they have been associated with.

We searched the PubMed database for articles in English. The main search terms included neurodegenerative diseases, Alzheimer’s disease, AD, Parkinson’s disease, PD, multiple sclerosis, MS, amyotrophic lateral sclerosis, ALS or spinocerebellar ataxia type 7, combined with competing endogenous RNA, ceRNA, long non-coding RNA, lncRNA, circular RNA, circRNA, pseudogene or miRNA. Since this is a relatively new research topic, no filter for the years was applied. We hand-searched the retrieved articles and selected the most relevant articles based on a subjective evaluation of their quality and relevance. Additional articles on specific topics were searched if needed.

### 2.1. ceRNA and Alzheimer’s Disease

In the context of NDDs, most ceRNA research has been done in AD, probably due to its growing prevalence in the aged population of most developed countries [118]. Such is the case that it is expected to affect 1 in 85 people worldwide by 2050 [119]. 

At molecular level, dementia associated with AD is characterized by proteinopathy with extracellular deposition of β-amyloid (Aβ) plaques in the brain and the presence of TAU neurofibrillary tangles (NFTs) in neurons. Consequences of AD at cellular level include deregulation of redox homeostasis and low-grade chronic inflammation [120]. 

#### 2.1.1. LncRNAs

As mentioned above, the deposition of Aβ plaques in the brain is one of the hallmarks of AD. β-secretase 1 (BACE1), one of the enzymes involved in the Aβ formation, is elevated in brains of AD patients [121,122]. Therefore, a misregulation of BACE1 may play an important role in this pathology. Both miRNAs and lncRNAs have been implicated in *BACE1* post-transcriptional regulation, one of the most relevant being the lncRNA BACE1-AS [24], an antisense transcript that regulates the expression of the homonym enzyme [123]. Also, BACE1-AS levels are increased in the plasma of AD patients and cell models [25]. Furthermore, Zheng and colleagues found that BACE1-AS shares many MREs with *BACE1* in a ceRNA regulation system that involves several miRNAs including miR-29, miR-107 and miR-124 [24]. Indeed, the authors demonstrated in vivo that BACE1-AS overexpression increased Aβ production in transgenic mouse brains. A subsequent computational study from available RNA-seq data published in the GEO database verified that levels of BACE1-AS and implicated miRNAs were also altered in AD clinical samples. Collectively, these data suggest that BACE1-AS acts as ceRNA to sequester miRNAs that target *BACE1*, thus preventing *BACE1* mRNA from degradation. This mechanism complements the one previously reported by which BACE1-AS increases the stability of *BACE1* mRNA through the formation of RNA duplexes [123]. BACE1-AS may also aggravate neurotoxicity in AD through other ceRNA mechanisms. It has been shown to sponge miR-214-3p and alter autophagy homeostasis. In fact, miR-214-3p levels are reduced in plasma from AD patients and in cell models [25], being this microRNA an inhibitor of autophagy and neuron apoptosis through transcriptional blockade of *Atg12* and having hence a neuroprotective effect [124]. 

In addition, BACE1-AS is a ceRNA for miR-132-3p [26], which is downregulated in AD patients [125] and has been shown to provide neuroprotection in the disease via modulating different target mRNAs and affecting multiple pathways, including regulation of synaptic proteins, tau phosphorylation and amyloid aggregation [126,127,128,129,130]. 

Besides BACE1-AS, other lncRNAs such as XIST (through miR-124), NEAT1 (which bind miR-124 and miR-107) and SOX21-AS1 (targeting miR-107) may regulate *BACE1* mRNA levels in AD cell and mouse models [27,29,30,31]. This highlights the complexity of *BACE1* regulation in AD being mediated by several ceRNA networks.

Other members of the miR-15/107 family can also regulate *BACE1* expression and the levels of other genes involved in AD [131]. In this line, NEAT1 along with two other lncRNAs, HOTAIR and MALAT1, bind miR-107, miR-103, miR-16, miR-195, miR-15a and miR-15b, exerting a regulatory effect on *CDK5R1* levels [32]. *CDK5R1* codes for the main activator of cyclin-dependent kinase 5 (CDK5), p35, essential for brain development and functioning so that its deregulation could be implicated in AD onset and progression [32]. Interestingly, in AD cellular models the expression of *CDK5* (along with *PTGS2* and *FOXQ1*) is regulated by miR-125b, which in turn is reversely regulated through MALAT1 [33]. Furthermore, MALAT seems to be involved in a third ceRNA network in AD through miR-30b and CNR1, a CNS-enriched cannabinoid receptor associated with learning and memory impairment, and significantly downregulated in AD [34].

Besides miR-124, the lncRNA XIST targets another miRNA and could participate in more than one ceRNET in AD. Wang et al. [28] showed that XIST knockdown inhibits Aβ protein fragment (Aβ25-35)-induced toxicity, oxidative stress and apoptosis in hippocampal neurons by binding miR-132, a miRNA widely reported in AD and known to target *SIRT1* [132,133]. Curiously, SIRT1 is involved in neuroinflammation and mitochondrial dysfunction in AD and regulates Aβ production through ROCK1 or ADAM10 [134,135,136,137]. In this line, Li et al. [35] reported that *ROCK1* is upregulated in AD cellular and mouse models and silencing of the lncRNA TUG1 depresses apoptosis of hippocampal neurons (like XIST1 knockdown) by elevating miR-15a and repressing *ROCK1* expression. ROCK1 is a ubiquitous serine/threonine kinase whose reduction has been reported to diminish Aβ levels by enhancing APP protein degradation in AD [138].

Aβ peptides have also been related to other lncRNAs that have been proposed to act as ceRNA in AD: the lncRNAs SNHG1, lncRNA-ATB, LINC00094, MIAT1 and Rpph1. Small nucleolar RNA host gene 1 (SNHG1) is a recently described lncRNA involved in the development of multiple human tumors, as well as in other types of diseases such as AD or PD [36,37,54,55,56,57,58,139,140,141,142,143,144]. In AD in vitro models, SNHG1 is upregulated and acts as a ceRNA for miR-137, regulating *KREMEN1* levels [36] and as a miR-361-3p sponge, modulating *ZNF217* [37]. KREMEN1 is a Wnt antagonist that also has pro-apoptotic effects in cells in a Wnt-independent manner [145,146], whereas *ZNF217* (*zinc finger gene 217*) has been described as a potential oncogene [147]. Notably, both studies reported that SNHG1 silencing partially reversed cell injury induced by Aβ25-35. Similar to SNHG1, suppression of the lncRNA-ATB protects cells against Aβ25-35-induced neurotoxicity by modulating *ZNF217* [38]. Nevertheless, this lncRNA may not act by regulation of miR-361-3p, but through miR-200.

LncRNA LINC00094 (also known as BRD3OS) has been reported to regulate blood-brain barrier (BBB) permeability in AD microenvironment by sponging miR-224-5p and miR-497-5p, both of which target *SH3GL2* mRNA [39]. *SH3GL2* codes Endophilin-1, an endocytosis protein markedly increased in the AD brain and involved in Aβ induced postsynaptic dysfunction [148]. Taken that miR-107 has been shown to protect from Aβ-induced BBB disruption and endothelial cell dysfunction by targeting *SH3GL2* mRNA [149], it seems possible that BACE1-AS, NEAT1, SOX21-AS1, HOTAIR and MALAT1 may also be involved in Aβ toxicity by regulating Endophilin-1 levels.

LncRNA MIAT (myocardial infarction associate transcript) is aberrantly expressed under neurovascular dysfunction [40], a condition that also aggravates AD pathogenesis by hindering Aβ clearance and, thus, increasing plaque levels in the brain [150]. In this sense, it has been proposed that MIAT modulates neural and vascular cell function and survival through the MIAT/miR-150-5p/*VEGF* axis, acting as a vascular dysfunction regulator [40,151,152]. Based on these premises, Jiang et al. studied the effect of MIAT knockdown in vivo [40] and observed a decrease in the number of cerebral microvessels, exacerbated neuronal loss, brain β-amyloidosis and neurodegeneration in mice, as well as behavioral deficits with significant impairment in the spatial learning capacity and memory. Thus, this work demonstrated the lncRNA MIAT involvement in the maintenance of adequate microvascular and neuronal function. 

In contrast to the lncRNAs discussed above, Rpph1 (ribonuclease P RNA component H1) seems to exert a neuroprotective compensation mechanism in AD pathology through three different ceRNA axes: Rpph1/miR-326/*PKM2* [41], Rpph1/miR-122/*Wnt1* [42] and Rpph1/miR-330-5p/*CD42* [43]. In particular, it has been shown to attenuate Aβ25-35-induced endoplasmic reticulum stress, neuronal injury and apoptosis [41,42], as well as to promote hippocampal neuron dendritic spine formation [43]. Rpph1 was first identified from bioinformatic analysis of whole transcriptome and microRNA sequencing data from a 12-month-old APP/PS1 transgenic mouse model of AD [43], where it was found upregulated in the cortex. This study not only helped to identify Rpph1 as an AD-related ceRNA, but also to establish a whole lncRNA-miRNA-mRNA ceRNET including 4 lncRNAs (C030034L19Rik, Rpph1, A830012C17Rik and Gm15477), 5 miRNAs (miR-182-5p, miR-330-5p, miR-326-3p, miR-132-3p and miR-484), and 1082 mRNAs. It was the first AD-associated ceRNA network based on APP/PS1 mouse model and was found to be enriched in mRNAs related to AD-associated genes, various signaling pathways (MAPK, neurotrophin, insulin/IGF, ErbB), as well as in the regulation of actin cytoskeleton, adherent junction, axon guidance and long-term potentiation.

In continuation to this work, the same laboratory studied the RNA expression profile in cortex samples from the same mouse model at different ages to determine the distinct lncRNA-associated ceRNA network (LncACeNET) that could be participating in the progression of the disease depending on the stage [96]. In general terms, these networks were found to be mainly involved in the cytoskeleton, postsynaptic density, cell–cell adherens junction and dendrite. Interestingly, LncACeNETs differentially expressed in the early stage model (associated with AD pathophysiology) differed from those altered in the advanced stage (more related to AD development), although a few lncACeNETs seemed to be contributing to AD pathology throughout the disease progression. Of all axes raised, authors highlight a series of lncRNAs that were identified as mmu_miR-122-5p and mmu_miR-679-5p ceRNAs, which both target *Klf4* (upregulated at the early stage of the disease) and *Akap5* (downregulated at late-stage), respectively, and LNC_000033, which acts as a ceRNA of 5 miRNAs that regulate the expression of *Synpo*. *Klf4* and *Akap5* are two genes related to Aβ-induced neuroinflammation, and synaptic plasticity and memory, respectively. As for *Synpo*, it codes for synaptopodin, an essential protein for dendritic spine plasticity of the developing hippocampus that was found upregulated at early stage [96].

Similarly, LncACeNETs have also been constructed based on human samples [97,98]. Wang and coworkers [97] built the first ceRNA network associated with NFTs in AD patients. The network built contained 41 lncRNAs and 630 mRNAs. Of these lncRNAs, three stand out: AP000265.1, RP1-145M9.4 and KB1460A1.5, focused on NFT related biological processes, including JNK cascade, protein phosphorylation and formation and development of neural tube, neural crest cells and epithelial tube morphogenesis. Recently, a second LncACeNET was constructed helping to identify neuroinflammation-related biomarkers for AD [98], including the lncRNA CTB-89H12.4. This lncRNA was shown to competitively bind miR-155-5p and to be significantly co-expressed with a group of genes involved in tau phosphorylation and amyloid formation, intercellular signal transduction and synapse function in addition to neuroinflammation.

As CTB-89H12.4, two other lncRNAs have recently been implicated in AD pathogenesis through tau phosphorylation and neuroinflammation. First, Yan et al. [44] demonstrated in AD cellular and mouse models that linc00507 mediated tau protein hyperphosphorylation by the activation of the p25/p35/GSK3β signaling pathway through regulating *MAPT* and *TTBK1* by sponging miR-181c-5p. As mentioned above, p35 is the main activator of CDK5, a kinase implicated in AD onset and progression [32]. Interestingly, p25 is the cleavage product of p35 that is able to bind to and activate CDK5 and the stability of the p25-CDK5 complex is higher than that of p35-CDK5 [44,153,154]. Hence, linc00507 could also be involved in CDK5 activity like the lncRNAs NEAT1, HOTAIR and MALAT1, but in this case in a more indirect way. Second, Zhou et al. [45] showed that the lnc-ANRIL could regulate neuroinflammation in AD since its knockdown suppressed apoptosis and pro-inflammatory cytokines (TNF-α, Il-1β, Il-6 and Il-17) and promoted neurite outgrowth by targeting miR-125a in an AD cellular model.

#### 2.1.2. CircRNAs

LncRNAs are not the only ncRNAs proposed as AD-associated ceRNAs, as these also include circRNAs. One of the best-studied circRNAs is CDR1as, an antisense circular transcript of the Cerebellar Degeneration-Related protein 1 (CDR1), highly expressed in the brain [155]. This circRNA appears to stabilize the *CDR1* mRNA [156] and to act as a miR-7 sponge, thereby being also called ciRS-7 [81]. Interestingly, the levels of ciRS-7/CDR1as are decreased in the hippocampus and neocortex of AD patients, which results in excess ambient miR-7 that downregulates its target mRNAs, such as that of ubiquitin-conjugating enzyme *UBE2A* that is essential for Aβ clearance [46]. This was the first circRNA-miRNA-mRNA axis reported to be dysregulated in AD [46]. Nevertheless, ciRS-7 could be part of another ceRNA protective network in AD by promoting the expression of ubiquitin C-terminal hydrolase UCHL1 that mediates APP and BACE1 degradation in an NF-κB dependent manner [47]. Although *NF-κB p65* subunit mRNA levels were not affected by ciRS-7 here, other studies have shown that the expression of *p65* is downregulated by miR-7 [48,49]. Therefore, ciRS-7/miR-7/*NF-κB(p65)* could represent another potential axis in AD and it might be influenced by APP, since it may reduce the levels of ciRS-7 [47].

Recently, other circRNAs have been found altered in AD, but, unlike ciRS-7, they are less well known. Among these, circ_0000950 promotes neuronal apoptosis, enhances inflammatory cytokine levels and suppresses neurite outgrowth in two cellular models of AD by directly sponging miR-103 and increasing the mRNA expression of *PTGS2*, a pro-inflammatory gene reversely regulated by miR-103 [50]. Another circRNA altered in AD is circHDAC9, which is decreased in animal and cellular models of this NDD, leading to the hyperrepression of the mRNAs regulated by miR-138 such as *Sirt1*, since it has been proposed to act as a sponge for this miRNA. As previously mentioned, Sirt1 regulates Aβ production in AD through ROCK1 or ADAM10 [134,135,136,137]. In agreement with the results obtained in animal and cell models, lower levels of circHDAC9 were found in the serum of AD patients [51]. This could be explained by the fact that Aβ downregulates circHDAC9, which, in turn, increases miR-138 expression and leads to a decrease of *Sirt1* and *ADAM10* levels, thus mediating synaptic function and APP processing in AD [51]. In line with this, Zhang et al. [52] demonstrated that the 42-residue β-amyloid (Aβ42) also triggers a significant downregulation of circHDAC9 in human neuronal cells, acting in this case as a sponge of miR-142-5p. Furthermore, the neuroprotective drug berberine alleviated Aβ42-induced neuronal damage in this cellular model by up-regulating circHDAC9 [52]. Although the study did not establish an mRNA target for miR-142-5p, it revealed a novel circRNA-miRNA axis in AD and opened the door to elucidate the entire ceRNA network. One possibility is that miR-142-5p may affect PSD-95 (a major scaffold postsynaptic protein that has been reported to be downregulated in brains of AD patients) by regulating the expression of *AKAP5* and *DRD1*, which are known to interact with PSD-95 [157]. As previously mentioned, *AKAP5* is a gene related to synaptic plasticity and memory and is part of a LncACeNET in AD [96]. As for DRD1, it is a dopamine receptor whose dysregulation could contribute to synaptic injury in AD [158].

The rest of the circRNA-associated ceRNA networks (cirCeNET) described in AD have been identified by transcriptome profiling studies in different animal models and patient samples. In the brain of the senescence-accelerated mouse model (SAMP8), Zhang et al. [99] constructed two AD-related cirCeNETs with circRNAs, miRNAs and mRNAs found differentially expressed. Among these, two ceRNA axes were identified as most likely to be involved in AD pathogenesis. On one hand, six circRNAs were predicted to act as sponges for the miRNA let-7g-3p and regulate a non-histone chromatin protein Hmgb2, involved in the Aβ clearance through up-regulation of the low density lipoprotein receptor-related protein Lrp1 [159,160]. On the other hand, five circRNAs could target miR-122-5p and control *iodothyronine deiodinase Dio2* mRNA expression, which is reduced in AD patients and is related to the myelination process [99,161]. In brain samples from the APP Tg2576 mouse model, Lee et al. [100] identified four AD-related cirCeNETs at two different disease stages. No circRNAs were consistently dysregulated at both stages (except for the mmu_circ_29980), suggesting a high time dependence in the regulation of the circRNA expression. One example of ceRNA axis predicted in this study in 12-month-old brain is mmu_circRNA_37345/miR_335-3p/*SLY*. *SLY* encodes Silymarin, a natural Aβ aggregation inhibitor with a large potential for the treatment of AD [162,163,164].

Other studies carried out in animal models analyzed specific brain areas such as the hippocampus, cortex or pineal gland. In the hippocampus of Aβ1-42-induced AD rat model [101], Wang et al. predicted an AD-related cirCeNET with 140 circRNAs, 140 miRNAs and 20 mRNAs. For example, circ_101834 and circ_004690/miR-7a-5p/*Aqp3*, where *Aqp3* codes for an aquaporin expressed in astrocytes and neurons, but its role in AD remains scarcely investigated [165]. In hippocampal samples too, but from SAMP8 mice model, Huang et al. [102] established a putative ceRNA network with one of the most significantly dysregulated circRNAs (mmu_circRNA_017963). This circRNA was strongly related to autophagosome assembly, exocytosis, apoptotic process, transport and RNA splicing, and it might potentially interact with 5 miRNAs and 313 mRNAs. Interestingly, in this same mouse model and tissue, Huang et al. studied the AD-associated circRNAs profile after treatment with Panax notoginseng saponins (PNS), used in traditional Chinese medicine [103]. Seven circRNAs were found significantly altered, and ceRNA networks were predicted for two circRNAs (mmu_circRNA_013636 and mmu_circRNA_012180). One example of ceRNA axis predicted was mmu_circRNA_012180/mmu_miRNA_6972-5p/*Gsdmd*. GSDMD is a key executive protein of pyroptosis, a highly inflammatory form of programmed cell death that has been reported to be an AD therapeutic target [166]. Thus, this axis could help to understand the underlying mechanism of action of PNS, which has been speculated to be a multi-targeted agent with anti-inflammatory properties [167]. In the cortex of APP/PS1 mice, Ma et al. [104] constructed five cirCeNETs. Among them, four were highlighted. Six circRNAs could bind to miR-466b-5p and regulate the expression of Scube2, an epidermal growth factor involved in neural development [168]. Four miRNAs may target *Sorbs2*, a gene that influences memory and dendritic development and its variants consistently were associated with delayed onset in AD [169,170]. Seven circRNAs could sponge miR-122b-3p and control the expression of *Cntnap2*, involved in the development of neural systems critical for learning and cross-modal integration [171]. Mmu_circ_0044900 could be a ceRNA for four miRNA that target *Creb* mRNA. CREB is an important transcriptional factor in the regulation of brain-derived neurotrophic factor (BDNF) and with the CREB-BDNF signaling pathway modulating cognitive status and Aβ toxicity in AD [172]. In the pineal gland of 5xFAD mice, Nam et al. [105] constructed a circRNA-miRNA network with the 10 circRNAs whose expression differences were more significant or whose expression levels were high. From it, a complete ceRNET was established, where circMboat2 and circNlrp5-ps could sponge miR-483 and regulate the expression of *Aanat*, a critical enzyme for melatonin synthesis [173].

In AD patients, three studies have provided circRNA expression profiling in the brain, and two in body fluids. Using different brain regions, Zhang et al. [106] constructed an AD-related cirCeNET with 276 circRNAs, 14 miRNAs and 1117 mRNAs. KIAA1586 ranked first within the AD risk circRNA-associated ceRNAs. This circRNA was predicted to competitively bind to 3 miRNAs (hsa-miR-29b, hsa-miR-101 and hsa-miR-15a) and regulate 159 mRNAs (some of them from AD-risk genes). Examples of ceRNA axes extracted from this network are KIAA1586/hsa-miR-15a/*PSEN2*, KIAA1586/hsa-miR-101/*UBE2A* and KIAA1586/hsa-miR-15a, hsa-miR-29b/*BACE1*. Accordingly, other studies have demonstrated the relationship between miR-29b and *BACE1* in AD [174,175], and the implication of miR-101 and miR-15a in the pathogenesis of the disease through the regulation of other AD-risk genes, such as *APP* [174,176].

In the brain cortex of patients, Dube et al. [107] identified AD-associated circRNAs through meta-analysis and generated a circRNA and linear mRNA co-expression network in order to infer the biological and pathological relevance of these circRNAs based on the linear transcripts they co-expressed with. circHOMER1 and circCORO1C were pointed out as prognostic and diagnostic biomarkers in AD. CircHOMER1 is derived from *HOMER1*, a gene involved in Aβ processing whose dysregulation may underlie the early phase of memory loss that occurs in AD [177,178,179]. This circRNA was predicted to have five binding sites for miR-651, which may target the AD-related genes *PSEN1* and *PSEN2*. Likewise, circCORO1C was predicted to contain two binding sites for miR-105 that target *APP* and *SNCA*. 

Similar to the last study, Lo et al. [108] investigated the circRNA profiles at different AD stages in four brain regions and constructed a circRNA-mRNA co-expressed network in the parahippocampal gyrus. In this net two circRNAs (hsa_circ_0000994 and hsa_circ_0005232) are originated from *SLC8A1*, a gene that codes for Na+/Ca2+ Exchange Protein 1 (NCX1) which could exert a neuroprotective role in AD [180]. Both circRNAs were predicted to be co-expressed with *EPHA4*, an emerging Aβ oligomer receptor that could be involved in the synaptic spine alterations found in AD [181]. However, Lo et al. did not predict miRNA binding sites in these circRNAs that could explain this co-expression through a competing endogenous mechanism. Future studies will be needed to elucidate common MREs in these circRNAS and *EPHA4* mRNA and confirm this ceRNA network.

In cerebrospinal fluid (CSF) from patients, Li et al. [109] constructed a circRNA-miRNA network with the top five up- and down-regulated circRNAs. Although this study did not predict mRNA targets, it pointed out two potential ceRNA networks. circ-TTC39C could sponge miR-210-3p, which induces dopaminergic neuron damage by reducing *BDNF* [182]. In turn, circ-PCCA may bind to miR-138-5p, which promotes tau phosphorylation by targeting *retinoic acid receptor alpha* (*RARA*) involved in the regulation of GSK-3β activity [183]. Another possible axis is circ-PCCA/miR-138-5p/*SIRT1* since miR-138-5p could also bind to *SIRT1* and mediate APP processing in AD (see above) [51]. This study also explored the clinical value of these circRNAs, and three of them (circ-AXL, circ-GPHN and circ-PCCA) could be potential biomarkers for predicting disease risk, guiding management and decision-making of AD [109].

In peripheral blood mononuclear cells (PBMCs) of patients, Li et al. [110] constructed a circRNA-miRNA network with the top 10 up- and down-regulated circRNAs, the ceRNA network of hsa_circ_082547 (one of the upregulated circRNAs, which could bind to more than 100 miRNAs and regulate 4 mRNAs), a ceRNA network with 3 circRNAs, 15 miRNAs and 223 mRNAs and a ceRNA network with 4 circRNAs, 20 miRNAs and 576 mRNAs. These networks were predicted to be strongly associated with inflammation, metabolism, and immune responses, which are all AD risk factors. Some examples of ceRNA axes potentially involved in AD pathogenesis are hsa_circ_061346/hsa-miR-5916-3p/APP, hsa_circ_000843/hsa-miR-335-3p/*SLC8A1* (involved in neuroprotection) and hsa_circ_061346/hsa-miR-103a-2-5p/*HOMER1* (involved in Aβ processing). Interestingly, this study also found that the differentially expressed circRNAs partially overlapped in this network analysis. In this sense, 13 circRNAs contain MRE sequences for hsa-miR-455-3p, which has been confirmed to bind at the 3’UTR of the *APP* gene and regulates its expression, exerting a protective effect on AD [184].

### 2.2. ceRNA and Parkinson’s Disease

PD is the second most frequent neurodegenerative disease after AD, and the most prevalent movement disorder, affecting approximately 1% of the population aged over 60 [185]. The etiology of PD is not fully understood, with age being the main risk factor. Clinically, it is characterized by bradykinesia (slow movement and impaired ability to move), rest tremor, muscle rigidity and postural instability [186]. At molecular level, these symptoms are caused by the loss of dopaminergic neurons of the substantia nigra pars compacta (SNpc) and the consequent loss of dopamine levels in the striatum [187]. In addition to this neuronal loss, PD is characterized by the presence of ubiquitinated cytoplasmic protein inclusions in the neurons located in the affected areas of the brain. These inclusions, known as Lewy bodies, are mainly composed of α-synuclein, product of the *SNCA* gene [188].

#### 2.2.1. Pseudogenes and lncRNAs

The first ceRNA network described in PD involved the regulation of *GBA* gene and its 96% homology pseudogene *GBAP1* by miR-22-3p binding [53]. *GBA* encodes a lysosomal glucocerebrosidase and its mutations represent the major genetic predisposing factor for PD [189]. Moreover, *GBA* has been closely related to key processes in this disease, including α-synuclein aggregation, lysosomal and autophagy dysfunction and endoplasmic reticulum stress [190]. However, this is not the best studied ceRNA in Parkinson’s disease. 

The lncRNA small nucleolar RNA host gene SNHG1 has been found upregulated in in vitro models of PD from neurons and microglia, as well as murine models [54,55,56,57,58], and it seems to contribute to PD pathogenesis through several complementary ceRNA mechanisms. It decreases viability and increases apoptosis in neurons by inhibiting miR-153-3p via sponging, which in turn regulates *PTEN* expression [54]. PTEN is an endogenous inhibitor of the PI3K/AKT/mTOR signaling pathway that has been previously demonstrated to be implicated in PD progression [191]. This is consistent with the fact that the PI3K/AKT/mTOR signaling pathway has been shown to prevent PD by promoting the survival and growth of dopaminergic neurons [192] and that miR-153-3p mediates neuroprotective effects against MPP^+^-induced cytotoxicity. It is worth noting that this signaling pathway has been also associated with the HAGLROS/miR-100/*ATG10* LncACeNET [59]. Additionally, SNHG1 promotes α-synuclein aggregation and toxicity by miR-15-5p binding and activating SIAH1 [56], an E3 ubiquitin ligase that promotes α-synuclein aggregation and apoptotic neuronal death [193,194]. miR-15-5p also targets the *GSKβ3* gene, establishing a new ceRNA axis implicated in neuron cytotoxicity and reactive oxygen species production [55]. Furthermore, SNHG1 contributes to neuronal death by competitively binding the miR-221/222 cluster and indirectly enhancing the expression of p27/mTOR, thus impairing autophagy [58]. On another note, this lncRNA also promotes microglial activation and neuroinflammation, aggravating PD pathology, through the SNGH1/miR-7/*NLRP3* axis [57]. Accordingly, SNHG1 silencing has been shown to promote autophagy [58], attenuate microglial activation and reduce dopaminergic neuron loss in the SNpc of PD mice [57]. 

LncRNA HOTAIR is overexpressed in both in vitro and in vivo models of PD [60,61,195]. This lncRNA was shown to induce neuronal injury by sponging miR-874-5p and stimulating *ATG10* [60], a gene also regulated by the HAGLROS/miR-100 axis [59] that codes for an enzyme essential for autophagosome formation and, therefore, autophagy. Moreover, HOTAIR positively regulates the expression of leucine-rich repeat kinase LRRK2 [195], whose mutations have been linked to both genetic and sporadic forms of PD [196]. HOTAIR knockdown protects against neuronal apoptosis in a PD cell model by repressing caspase 3 activity [195]. This lncRNA also regulates RAB3IP, an important activator of Rab proteins, via miR-126-5p sponging in PD models [61]. The role of RAB3IP in PD is still unknown, although it has been reported to regulate neurite outgrowth and spinal development [197,198]. In this sense, the HOTAIR/miR-126-5p/*RAB3IP* axis could provide a new pathomechanism and therapeutic target in the disease. To unravel the role of HOTAIR in vivo, PD mice were injected with HOTAIR shRNA, resulting in protection against the initiation and development of PD, including cognitive impairment and bradykinesia. At molecular level, these mice had lower levels of *RAB3IP* and increased miR-126-5p expression compared to the PD control group. Moreover, at cellular level, shRNA-HOTAIR increased the number of TH-positive cells, reduced α-synuclein-positive cells and protected neurons from apoptosis [61]. 

*RAB3IP* is also a part of another ceRNA axis, in this case together with the lncRNA NEAT1 and miR-212-5p [62]. This axis was found unbalanced in a cell model of PD, with NEAT1 and *RAB3IP* being upregulated and miR-212-5p downregulated. Importantly, NEAT1 is increased in peripheral blood cells of PD patients [199]. Its knockdown or miR-212-5p overexpression in vivo suppresses neuronal apoptosis, inflammation and cytotoxicity [62], supporting the observations that RAB3IP overexpression is detrimental in PD models [61]. NEAT1 also enhances apoptosis, inflammation and cytotoxicity in PD models through two other ceRNA mechanisms, specifically through NEAT1/miR-1277-5p/*ARHGAP26* [63] and NEAT1/miR-124 [64] axes.

LncRNAs AL049437, MALAT1, SNHG14, lincRNA-p21, GAS5 and BDNF-AS, upregulated in mouse and in vitro models of PD, have also shown to exert a detrimental role and contribute to the progression of the disease [65,67,70]. AL049437 was demonstrated to act as a regulatory mechanism for the mitogen-activated protein kinase MAPK1 [65], a kinase that has been related to autophagy via miR-205-5p sponging [200,201,202]. AL049437 silencing mitigated neuronal injury in vitro, increasing neuronal viability, reducing cell apoptosis and alleviating neuroinflammation and oxidative stress, and this was reverted by miR-205-5p silencing or *MAPK1* overexpression. Likewise, MALAT1 also acts through a ceRNA mechanism involving miR-205-5p but to regulate *LRRK2* expression, contributing to cell apoptosis in in vitro and in vivo models of PD [66].

Taken together, these data may point towards a common regulatory network where all MALAT1, AL049437 and HOTAIR, which also regulated *LRRK2* [195], sponge miR-205-5p, resulting in the post-transcriptional regulation of the *MAPK1* and *LRRK2* genes.

Additionally, Liu et al. proposed a second ceRNA mechanism, in which MALAT1 would serve as a molecular sponge of miR-124, although the mRNA downstream this edge was not investigated [67]. Later studies by Lu and colleagues suggest that the downstream piece of this axis may be *DAPK1* [68], which encodes for a protein kinase that intervenes in apoptosis and autophagy regulation [203,204]. In this sense, MALAT1 knockdown led to increased miR-124-3p levels and *DAPK1* downregulation, alleviating cell apoptosis in vitro and in vivo and improving behavioral changes in PD mice. Alternatively, other mRNA targets of miR-124-3p are likely to play a role in PD. Those include *MAP3K3*, *RELA/p65* [205], *Bim* [206], *SQSTM1*, *MAPK11* [207], *STAT3* [208], and *ANXA5* [209], among others [210,211,212,213]. Lastly, MALAT1 was also shown to directly regulate α-synuclein expression via targeting miR-129 [69]. Similar to MALAT1, SNHG14 and lincRNA-p21 modulate α-synuclein levels through miR-133 and miR-1277-5p sponging, respectively [70,71], and inhibition of these two lncRNAs mitigated dopaminergic neuron injury in vitro and in vivo [69,70,71]. LincRNA-p21 also acts at the level of neuroinflammation and microglial activation via the lncRNA p21/miR-181/*PRKCD* (PKC-δ) family feedback loop [72], and plays a role in dopaminergic neuron death by increasing oxidative stress and neuroinflammation through the lincRNA-p21/miR-625/*TRPM2* axis [73]. Finally, GAS5 and BDNF-AS have been shown to regulate *NLRP3* expression via competitive sponging of miR-223-3p and miR-125b-5p, enhancing in vivo and in vitro microglial inflammatory response [74] and promoting neuronal apoptosis [75], respectively.

As opposed to the above-mentioned ceRNA networks, lncRNAs can also have a protective role in PD. Such is the case of the lncRNA Mirt2 or lncRNA H19. Mirt2 prevents TNFα-triggered inflammation via the repression of miR-101 [76], whereas lncRNA H19 protects against dopaminergic neuron loss in PD mice via regulating miR-301-3p/*HPRT1* [77], and consequently Wnt/β-catenin signaling pathway. LncRNA H19 can also act through miR-585-3p/*PIK3R3* axis [78], being *PIK3R3* a gene that codes for a regulatory subunit of PI3K associated with increased susceptibility to PD [214].

Besides experimentally validated ceRNA networks, in recent years, bioinformatic predictions have uncovered new disease-associated ceRNAs and their principal mechanisms of action. In 2017, Lin and colleagues [111] established the first LncACeNET from a synthetic cellular model of PD, including three lncRNAs (AC009365.4, RPS14P3, G046036). The most prominent mRNAs involved in these networks were *IRF1* and *RIMKLA* (potentially regulated by AC009365.4), *NAV1* (related to RPS14P3) and *SACS* and *SDC2* (G046036), all thought to be related to neurodegenerative processes although their role in PD still awaits clarification. One year later, Chi et al. established a PD-associated network from differentially expressed RNAs in PD patients’ blood versus controls [112]. This network consisted of 7 lncRNAs (including XIST, PART1, MCF2L2, NOP14-AS1, LINC00328, LINC00302 and FAM215A), 3 miRNAs (miR-7, miR-133b and miR-433) and 55 mRNAs especially enriched in the GnRH, insulin and MAPK signaling pathways. More recently, Zhang and colleagues [113] identified a new network in a substantia nigra array from PD patients and matched healthy controls that comprised 9 lncRNAs, 18 miRNAs, and 185 mRNAs functionally related to autophagy, DNA repair and vesicle transport, all critical cellular processes in PD. Based on the most significant relationships, they established a second ceRNA network that was validated using external data. It included four lncRNAs (KCNQ1OT1, LINC00467Z, SOX2-OT and NEAT1), nine miRNAs (miR-3163, miR-424-5p, miR-215-5p, miR-193-3p, miR-195-5p, miR-1-3p, miR-92b-3p, miR-520g-3p and miR-124-3p) and six mRNAs (*PTBP1, FBXL7, SRSF1, PTBP2, UBE2Q2* and *RBBP6*) related to mRNA metabolism, mitochondrial function and injury, DNA damage and protein polyubiquitination. Interestingly, *PTBP1* was previously reported in PD [215,216].

#### 2.2.2. CircRNAs

As previously mentioned, miR-7 is part of a PD-related ceRNET where the lncRNA SNHG1 and *NLRP3* mRNA could compete for it [57]. However, miR-7 is also capable of binding to *SNCA* mRNA and repress α-synuclein protein levels [79]. Interestingly, a decrease of miR-7 levels was detected in PD models (MPTP treated mice and MPP+ treated cells) that could contribute to the pathogenesis of the disease [79]. Since CDR1as/ciRS-7 contains over 70 binding sites for miR-7 and has been found to be altered in other NDDs such as AD [46,47] (see above), ciRS-7/miR-7/*SNCA* axis has been proposed as a possible ceRNET in PD [80,81,82,83]. However, it is not known if the expression of ciRS-7 in PD-affected tissues is altered. What has been experimentally demonstrated [84], though, is that a circRNA from *SNCA* gene itself (hsa_circ_0127305, also called circSNCA) acts as a ceRNA of miR-7 and upregulates *SNCA* in PD. Furthermore, pramipexol (PPX), a common treatment for PD, exerts suppressive effects on circSNCA expression. In agreement, Sang and coworkers showed that the inhibition of circSNCA and SNCA reduce apoptosis and promote autophagy, thus attenuating the progression of PD [84]. In light of these findings, miR-7 could be involved in a third ceRNET in PD (circSNCA/miR-7/*SNCA*), or even ciRS-7, circSNCA and SNHG1 may be part of the same ceRNET (miR-7/*SNCA* and *NLRP3*) exerting a cooperative regulation. 

In contrast to circSNCA, the circRNAs circzip-2 and circDLGAP4 have been found altered in PD with a protective role. On one hand, Kumar et al. [85] observed in a transgenic *C. elegans* PD model a significant down-regulation of circzip-2, a circRNA synthesized from *zip-2* gene whose human ortholog codes for CCAAT-enhancer-binding protein (C/EBP), a bZIP transcription factor involved in PD by regulating α-synuclein levels [85,217]. Through bioinformatic analysis, it was predicted that circzip-2 could sponge miR-60. Hence, a decrease of circzip-2 may enhance the miR-60 activity, which leads to downregulation of protective genes, including *M60.4*, *ZK470.2*, *igeg-2* and *idhg-1*, the ortholog of mitochondrial isocitrate hydrogenase (NAD+). On the other hand, Feng et al. [86] described in PD models (MPTP-induced mice and MPP+-induced cells) a decreased expression of circDLGAP4, which demonstrated attenuate the neurotoxic effects in vitro. The authors predicted that miR-134-5p could be a target of circDLGAP4 both in human and mouse and, in agreement, found this miRNA upregulated in both models. Finally, the same study demonstrated that circDLGAP4/miR-134-5p axis regulates CREB signaling, as well as the transcription of CREB downstream target genes *BDNF*, *Bcl-2* and *PGC-1a*, all of which are neuroprotective factors involved in many NDDs including AD and PD [218,219,220].

Recently, two studies have provided circRNA expression profiling in PD brain. Jia et al. [114] identified circRNAs differentially expressed in four brain regions of MPTP mice and constructed a ceRNA network with 6 of these circRNAs and its prediction downstream targets that could be involved in the PD-related processes (13 miRNAs and 112 mRNAs). Among them, two circRNAs were highlighted. Mmu_circ_0003292 could act as a sponge of miR-132 and upregulate the expression of Nr4a2, an important transcription factor with a neuroprotective effect in the pathogenesis of NDDs (including PD, AD and MS) [221]. Mmu_circ_0001320 was predicted to sponge miR-124 and regulate the expression of Sox9, a positive regulator of astrogliosis, this miRNA-mRNA interaction being previously reported in PD models [222].

Hanan et al. [115] explored the expression patterns of circRNAs in three different brain regions from PD patients. Unlike the rest of transcriptome profiling studies, this research did not construct ceRNETs with the circRNAs identified. However, seeking differentially expressed circRNAs in all brain tissues from PD and healthy individuals a total of 24 were identified, among them stood out a significant increase of circSLC8A1. This circRNA is derived from the *SLC8A1* gene that codes for NCX1, which could be involved in AD (see above) and also PD [223]. circSLC8A1 was significantly upregulated in the substantia nigra of AD patients, was directly modulated by oxidative stress and carries seven binding sites for miR-128. Accordingly, an increase of miR-128 mRNA targets (*BMI1*, *SIRT1* and *AXIN1*) was reported in PD brains. Therefore, a new ceRNET (circSLC8A1/miR-128/*BMI1*, *SIRT1*, and *AXIN1*) was described that could be involved in PD pathology through oxidative stress [115]. 

### 2.3. ceRNA and Multiple Sclerosis

MS is a chronic autoimmune disease, characterized by an immune-cell-mediated attack to the white matter in the CNS causing inflammation, demyelination and axonal loss [224,225]. MS onset usually occurs between 20 and 40 years of age, and results in a life-long physical and cognitive disability [226]. Although MS lacks a preclinical mouse model that could faithfully recapitulate the disease progression, experimental autoimmune encephalomyelitis (EAE) mice are widely used as MS models to investigate and to test therapies targeting the inflammation component.

Lately, a large number of studies support the importance of lncRNAs role in the differentiation, function and misproportion of immune cells [87], as well as autoimmunity and human inflammatory response [227,228], suggesting that they may play a pivotal role in MS. Out of the different classes of immune cells, there is increasing evidence on the pathogenic role of T helper 17 (Th17) cells and the imbalance between these and regulatory T cells in various autoimmune diseases such as MS and neuromyelitis optica [229]. Two lncRNAs (Gm15575 and PVT1) with a ceRNA mechanism of action have been shown to affect the function of Th17 in MS. On one hand, lncRNA Gm15575, which is enriched in Th17 cells and the spleen of EAE mice, was reported to act as a miR-686 sponge. As a consequence, Gm15575 positively regulated the expression of CCL7 [87], a proinflammatory chemokine highly expressed in Th17 cells that in MS patients promotes the infiltration of inflammatory cells into the CNS, stimulating the advance of the disease [230]. This study also showed that Gm15575 promoted IL17 expression and that silencing this lncRNA led to a reduction in *IL17* mRNA and protein levels, as well as mRNA of *RORϒt*, a lineage defining transcription factor required for Th17 cell differentiation and function [87]. These authors postulate that CCL7 secreted by Th17 cells could play a role in promoting the production of other proinflammatory cytokines (such as IL17) and differentiation of CD4+ lymphocytes to Th17, given RORϒt function [231]. This, together with its power to recruit immune cells, would aggravate the disease [87]. On the other hand, lncRNA PVT1 is downregulated in MS patients [232] and in the spinal cord of EAE mice [88]. Wu et al. demonstrated that it acts as a ceRNA for miR-21-5p, thus regulating the expression of SOCS5 [88], a protein from the suppressor of cytokine signaling (SOCS) family, which is also downregulated in MS patients [233]. SOCSs proteins are rapidly transcribed in response to intracellular JAK-STAT signaling [234], regulating the cytokine-induced immune response and therefore playing an important for the progression of multiple sclerosis [233]. Moreover, low levels of PVT1 and *SOCS5* in EAE mice were associated with an increased number of Th17 cells and markers of inflammation (IL-17, IL-6, IL-1β and TNF-α) in the spinal cord of this model. These inflammation markers were reduced with the administration of exosomes from M2 microglia, which are enriched in PVT1; and this effect was reversed by the administration of M2 exosomes with shRNA for PVT1. Hence, it was proposed that PVT1 could ultimately inhibit the proinflammatory response of Th17 cells through a JAKs/STAT3-mediated pathway. 

The above results indicate that the lncRNA-ceRNA networks influence the inflammatory response in MS. This idea is also supported by the TUG1/miR-9-5p/*NFkB1(p50)* ceRNET described by Yue et al. in 2019 [89]. TUG1, or lncRNA taurine-upregulated gene 1, was first reported as upregulated in the serum of MS patients by Santoro et al. [227] and last year was proven to regulate p50 through miR-9-5p sponging [89]. Importantly, TUG1 down-regulation in vivo improved mouse behavior, decreased the levels of pro-inflammatory cytokines such as TNF-α, IFN-γ, IL-6 and IL-17, and increased the anti-inflammatory cytokine IL-10 in EAE mice.

Besides inflammation, demyelination is another important MS hallmark, a process that could also be influenced by lncRNAs acting as miRNAs sponges. Sulfasalazine, a drug that promotes remyelination and improves the outcome of MS patients [235,236], exerts its beneficial effect at least partially through the inhibition of the lncRNA HOTAIR. This lncRNA acts as a sponge for miR-136-5p, promoting AKT2-mediated NF-kB activation, thus favoring the microglial shift toward a proinflammatory M1-like phenotype [90], detrimental in MS. Moreover, lncRNA GAS5 has been proposed as a ceRNA for miR-137 participating in the demyelination process [91] based on the following premises: (i) GAS5 is upregulated in MS patients while miR-137 is downregulated, (ii) GAS5 acts as a miR-137 molecular sponge in ischemic stroke [237] and (iii) GAS5 exacerbates demyelination and inhibits microglial M2 polarization in EAE mice and human primary cell culture [238]. Based on their results, they also proposed serum GAS5 and miR-137 as MS biomarkers for negative prediction and severity, respectively [91].

Finally, lncRNA MALAT1, which is also dysregulated in MS, can cause alternative splicing abnormalities of MS-associated genes (e.g., *IL7R*, *SP140*) and contribute to back-splicing of approximately 50 circRNAs [239]. Since this lncRNA has been pointed out as a potential MS biomarker [240] and is involved in ceRNETs in other NDDs (such as AD or PD, see above), it is not surprising that MALAT1 could be also involved in a ceRNA axis in MS.

In contrast to lncRNAs, no circRNA-associated ceRNA networks implicated in MS has yet been validated. Nevertheless, novel isoforms and an upregulated circRNA (hsa_circ_0106803) from the *GSDMB* gene, associated with susceptibility to several autoimmune diseases and involved in pyroptosis [241], have been found in PMBCs of MS patients [92]. Several miRNAs were predicted to contain more than one target site in hsa_circ_0106803 and, among these, miR-1275 and miR-149 are differentially expressed in blood from MS patients [242,243]. It has been reported that miR-149 binds to *ASIC1a* and reduces its levels [244]. *ASIC1a* encodes a subunit of acid-sensing ion channel, which is overexpressed in acute MS lesions and could be implicated in the neuronal pathogenesis of this disease [245,246,247,248]. In light of this evidence, a ceRNA network has been proposed where hsa_circ_0106803 could modulate the progression of MS by regulating the expression of *ASIC1a* mRNA through miR-149 [93]. Another study in PBMCs of MS patients [94] found two downregulated circRNAs (hsa_circ_0005402 and hsa_circ_0035560/hsa_circ_0003452_2) from *ANXA2* gene, encoding an annexin related to immune-mediated diseases [249,250]. Through bioinformatic analysis, it was predicted that hsa_circ_0005402 presents a single binding site for 17 miRNAs and two binding sites for miR-1248 and miR-766. Curiously, hsa_circ_0005402 shares 14 common miRNA targets with hsa_circ_0035560, so a cooperative regulation of a circRNA-miRNA-mRNA axis could be involved in MS pathogenesis [93,94]. However, further studies are needed to elucidate all components of this axis and confirm this potential cooperative ceRNA network in MS. 

### 2.4. ceRNA and Amyotrophic Lateral Sclerosis

ALS is the third most common progressive neurodegenerative disease, affecting 150,000 people in the world [251]. It is characterized by atrophy of voluntary muscles and paralysis as a consequence of the progressive and selective loss of motor neurons in the spinal cord and brain [252]. ALS is a complex and multifactorial disease with many dysregulated cellular processes but, in recent years, alterations in genes associated with RNA metabolism have been in the spotlight. This is supported by the fact that two of the most important genes associated with this disease (*FUS* and *TARDBP*) are involved in transcription and RNA metabolism [253] and, notably, are able to regulate biogenesis or expression of certain lncRNAs and circRNAs [254,255,256].

In agreement, a potentially pathological paraspeckle enrichment was observed in motor neurons of ALS patients’ brains [257]. Paraspeckles are nuclear bodies formed by a set of specialized RNAs and proteins, among which are the RNA-binding proteins FUS and TDP-43, encoded by the *FUS* and *TARDBP* genes, respectively. These specialized RNAs are especially enriched in the lncRNA NEAT1_2, which has been identified predominantly expressed in spinal motor neurons in early phases of ALS pathological process and was shown to directly bind FUS and TDP-43 [257,258]. The frequency of paraspeckle formation is highly increased during the early phases of ALS course, so it has been proposed that NEAT1_2 could act as a scaffold of the RNA-binding proteins in the nucleus [259] by sequestering them and forming aggregates, thus playing a possible important role in RNA metabolism imbalance and ALS pathogenesis. Whether this lncRNA could also act through other pathways such as the ceRNA mechanism, already observed in a cellular PD model [62], remains to be determined. The lncRNA C9ORF72-AS has also recently been related to ALS, although its function in ALS is still unknown. However, it seems that the first exon of this lncRNA has different binding sites for miRNAs [260].

Whole transcriptome RNA-seq analyses have revealed differential expression of lncRNAs both in blood PBMCs and iPSC-derived motoneurons from ALS patients [254,261,262], although their possible mechanisms of action have not been addressed yet. Interestingly, at least some of the lncRNAs deregulated in human iPSC-derived motoneurons were conserved between mouse and human, concurring with those of mouse embryonic stem cells with the *FUS* mutation [254].

Similar to lncRNAs, little is known about circRNA in ALS. However, three circRNAs have shown some potential as diagnostic biomarkers [116]. Dolinar et al. [116] investigated the expression profile of circRNAs and identified 425 differentially expressed in leukocytes from sporadic ALS (SALS) patients. Among these, hsa_circ_0023919, hsa_circ_0063411 and hsa_circ_0088036 showed the highest significance as well as clinical relevance. Curiously, the expression levels of several circRNAs were positively correlated with each other, suggesting that they could be involved in similar biological processes and/or co-regulated [116]. Although possible circRNA interaction networks were not investigated, the authors predicted miRNA targets for two affected circRNAs. Hsa_circ_0023919 was found significantly downregulated in SALS patients and contains two binding sites for hsa-miR-9. Accordingly, two previous studies confirmed the upregulation of miR-9 in both mouse model of ALS [263] and in blood samples of ALS patients [264]. It has been reported that miR-9 directly binds to and reduces the mRNA levels of *NEFL* [265], which encodes the neurofilament light polypeptide. Interestingly, aggregation of intermediate filament is a characteristic ALS hallmark. In light of this evidence, one potential biomarker based on ceRNA hypothesis has been proposed in ALS (hsa_circ_0023919/miR-9/*NEFL*), where hsa_circ_0023919 could act as a miR-9 sponge and regulate the metabolism of intermediate filaments (NEFL) observed in ALS [117]. In contrast, hsa_circ_0063411 was found upregulated in patients. This circRNA contains one binding site for miR-647, which has been seen expressed in spinal cords from healthy subjects, but not from SALS patients [266]. However, the study did not establish an mRNA target of hsa_circ_0063411/miR-647 axis, so further studies are necessary to elucidate this potential ALS ceRNA network. One possibility is that hsa_circ_0063411 may bind to miR-647 and increase the expression of *PTEN*, a direct target of miR-647 [267]. As mentioned above, *PTEN* has been seen involved in AD- and PD-associated ceRNA networks. Since PTEN has been reported as a potential therapeutic target in motor neuron diseases, including ALS or SMA (spinal muscular atrophy) [268,269,270], it remains possible that *PTEN* mRNA could also be part of ALS-linked ceRNETs.

It is important to note that, although there are no experimentally confirmed ceRNAs associated with ALS, ceRNA mechanisms can regulate myogenesis [271], an altered process in ALS due to the relevant muscular component in this disease. These mechanisms include the lncRNAs linc-MD1 [272], lncRNA H19 [273], MALAT1 [274], lnc-mg [275], lncMD [276], Yam [277] and Sirt1-AS [278], among others.

### 2.5. ceRNA and Spinocerebellar Ataxia Type 7

Spinocerebellar Ataxia Type 7 (SCA7) is a rare inherited neurodegenerative disease caused by polyglutamine repeat expansion in Ataxin 7 (ATXN7), a product of the *SCA7* gene. This results in formation of protein aggregates and decreased protein activity [279] that contribute to neurodegeneration. The disease comprises a phenotypic spectrum ranging from adolescent- or adult-onset progressive cerebellar ataxia and cone-rod retinal dystrophy to infantile- or early-childhood-onset with multiorgan failure, accelerated course, and early death [280]. 

In SCA7, crosstalk between ncRNAs have been shown to contribute to cell-specific neurodegeneration. Specifically, lnc-SCA7, a highly conserved lncRNA derived from the retrotransposition of the *ATXN7L3* gene (a distant paralog of *SCA7*), has been shown to regulate the expression of the *ATXN7* gene in a miR-124-dependent manner [95]. Tan et al. found that lnc-SCA7 and *ATXN7* mRNA levels increased in both patient-derived fibroblasts and SCA7 mouse model brain, while miR-124 levels decreased. Furthermore, in mice, this increase was more prominent in tissues relevant to SCA7, such as the retina and cerebellum. Surprisingly, reduction of lnc-SCA7 levels led to a depletion of both mature and precursor miR-124 levels, generating a novel negative feedback loop involving *ATXN7* and miR-124. 

## 3. RNA Editing Alteration and ceRNA Networks in Neurodegenerative Diseases 

RNA editing is an important mechanism of post-transcriptional processing that can modify RNA molecules by altering its sequences through insertion, deletion, or conversion of a nucleotide [281,282]. Recent discoveries suggest that RNA editing critically regulates neurodevelopment and normal neuronal function, for which some crucial aspects of neurodegenerative diseases may stem from the modification of both coding and non-coding RNA [282,283,284]. 

The most common type of RNA editing is the conversion of adenosine to inosine (A-to-I), in which enzymes encoded by the adenosine deaminase acting on RNA (ADAR) gene family catalyze the deamination of adenosine (A) nucleotides to inosines (I) [285]. Critical consequences are derived from this modification, since inosine (I) is interpreted by the translation and splicing machineries as guanosine (G) [286]. Editing of pre-mRNA coding regions can lead to codon change that may result in increased diversity of protein isoforms and their respective function [281]. However, most of the RNA editing happens in non-coding RNAs, which can affect their stability, biogenesis and target recognition [281,286,287]. In fact, it has been reported that ADAR is involved in circRNA biogenesis by editing and destabilizing the flanking Alu repeat sequences, which makes circRNA production less favorable [287,288]. Moreover, editing events can affect both the maturation and the expression of miRNAs, but if the modification occurs in MREs or in miRNA seed regions (regions in miRNA sequence that largely determine the binding specificity on its targets), the spectrum of miRNA targets, or “targetome”, shall be changed [289]. Therefore, a single editing site in an RNA molecule could drastically modify its function, resulting in new or different ceRNA networks that regulate gene expression.

Interestingly, A-to-I editing has been reported specifically reduced in SALS motor neurons due to the progressive downregulation of ADAR2 [290,291]. Based on this evidence, Hosaka et al. [292] searched for extracellular RNAs with ADAR2-dependent A-to-I sites that may reflect the intracellular pathological process and thus could be potentially good ALS biomarkers. A total of six RNAs were identified. Among these, a circRNA (hsa_circ_0125620, also called circGRIA2) with an ADAR2-dependent site was detected in human SH-SY5Y neuroblastoma cells as well as in their culture medium [292]. Therefore, variations in RNA editing efficiency in ALS, as a consequence of decreased ADAR2 activity, could be potentially measured in peripheral circRNAs and other relatively stable ncRNAs. In light of this evidence, this editing phenomenon may be considered a very important aspect, since it allows obtain relevant information of disease pathological process from non-coding RNAs.

Other NDDs, such as AD and PD, also present alterations in RNA editing patterns [115,283,293,294]. In fact, a recent study has explored how RNA editing in AD contributes to the regulation of AD-related processes in blood cells in two populations of patients [295]. Results identified differentially edited sites predicted to disrupt miRNA target sites in five genes. In all cases, decreased editing was observed in AD suggesting a greater miRNA-binding affinity relative to controls [295]. In light of this evidence, alterations in RNA editing could result in a specific RNA profile, given by different amount of RNAs, modified interaction networks and editing levels or efficiencies changes in A-to-I sites, that could be useful to identify new robust biomarkers of these NDDs (Figure 2).

## 4. Conclusions and Future Perspectives

The vast majority of NDDs can be definitively diagnosed only after death or in advanced stage, and their previous diagnosis is based on ruling out other possible causes for the symptoms. For most NDDs, there is no cure or treatment capable of reversing the damage due to neuronal death. Therefore, it is critical to find new biomarkers that would facilitate an early diagnosis, prognosis and efficient monitoring of therapeutic interventions.

In the search for new biomarkers, non-coding RNAs have been proposed as promising tools for diagnosis and prognosis. Many ncRNAs often arise from genes that cause NDDs or are somehow involved in the development of one of these disorders (like BACE1-AS or circSNCA). Thus, ceRNETs established by these ncRNAs could well be, at least in some cases, disease and even stage-specific. However, as reported in this review, ncRNAs are commonly misregulated in several NDDs (Figure 3). This is the case, for example, of the lncRNAs SNHG1 and HOTAIR, which are altered in AD [36,37] and PD [54,55,56,57,58], and PD [60,61] and MS [90], respectively. However, their miRNA targets may vary depending on cell types affected by the disease and, therefore, the mechanism of action may also differ. Similarly, miR-7 has been shown sponged by ciRS-7/CDR1as and circSNCA in AD [46,47,48,49] and PD [84], respectively, being detrimental in the first case and beneficial in the second, due to regulation of different target mRNAs. The apparent discrepancy between the anti and pro cell death activity of miR-7 reflects the complex regulatory role of miRNAs, so further research is required to clarify their function in different cellular and disease contexts.

In this way, by analyzing various elements of the altered ceRNETs, it may be possible to differentiate one NDD from another even if there were common components. Ideally, working with several correlatable molecular targets at the same time (lncRNAs/circRNAs/pseudogenes-miRNA-mRNAs) increases the sensitivity and reliability of ceRNETs as biomarkers. It should be noted that ceRNETs construction also contributes to the identification of new molecular mechanisms of gene regulation that may lead to a better understanding of the etiopathogenesis of the diverse NDDs, as well as to reveal new therapeutic targets and obtain relevant information about the pathological processes of the disease.

In this sense, ceRNETs may also reflect the editing efficiencies of ADAR, a post-transcriptional phenomenon dysregulated in several NDDs. RNA editing can affect the levels and the efficiency of RNA interaction networks, so its alterations could provide a specific RNA fingerprint that helps in the diagnosis or prognosis of NDDs. Finally, the described crosstalk between the RNA molecules in certain ceRNETs is relatively conserved between species, paving the way for translation of data obtained from animal models into clinical practice [297,298]. 

Among the main advantages of ceRNETs for biomarker research, the fact that these ncRNAs are easily accessible is noteworthy, since they are extremely stable in circulation and may be detected in exosomes. Such is the case for circRNA CDR1as/ciRS-7 and lncRNA MALAT1, found in exosomes. Interestingly, levels of ciRS-7 in these vesicles depend on the intracellular abundance of the miRNA that it sponges (miR-7) [299]. Furthermore, ciRS-7 and MALAT1 may regulate miRNA expression in target cells after exosomal delivery modulating their phenotype, since these ceRNAs retain their biological activity [299,300]. Therefore, ciRS-7 and MALAT1 together with other circulating ncRNAs (e.g., NEAT1, GAS5, hsa_circ_061346, hsa_circ_000843) represent promising candidates for peripheral ceRNA biomarkers of NDDs. Although many of the ncRNAs discussed earlier have not been reported in exosomes to date, some of them are predicted to be detected in human blood exosomes by exoRBase (e.g., circSLC8A1, circCORO1C, SNHG1, BACE1-AS) [301]. Indeed, it has recently been demonstrated that plasma exosomal BACE1-AS levels could serve as a biomarker of AD [302,303]. 

Because ceRNA interaction networks are multifactorial, they may represent an advantage in studies of these complex neurodegenerative disorders, one being at the level of biomarkers (combined RNA biomarkers panels) and another at the level of therapeutic targets (modulate the levels of multiple disease-associated RNAs at once by just targeting one).

Nevertheless, it must be taken into account that there is still much to do, since these networks are very complex and their interactions must be experimentally defined [297]. In this sense, some “non-canonical” aspects of ncRNAs have also been described: (i) circRNAs that can also sponge or serve as a decoy for RBPs or lncRNAs, (ii) miRNAs that may increase the expression of target genes, (iii) lncRNAs that can be precursors of smaller ncRNAs and can regulate miRNA and circRNA biogenesis, (iv) miRNAs that can direct Ago2 to degrade lncRNA and circRNA, (v) lncRNAs that compete with miRNAs for the target site of mRNA, and (vi) context-specific miRNA function and target identification [304,305,306,307,308,309,310,311].

Although the full extent of ceRNA networks still needs to be still determined, the competition of ncRNA and mRNAs for miRNAs constitutes a key point of gene regulation that could underlie some pathological aspects of neurodegenerative diseases, favoring at the end the identification of specific pathological mechanisms for each disease.

## Figures and Tables

**Figure 1 ijms-21-09582-f001:**
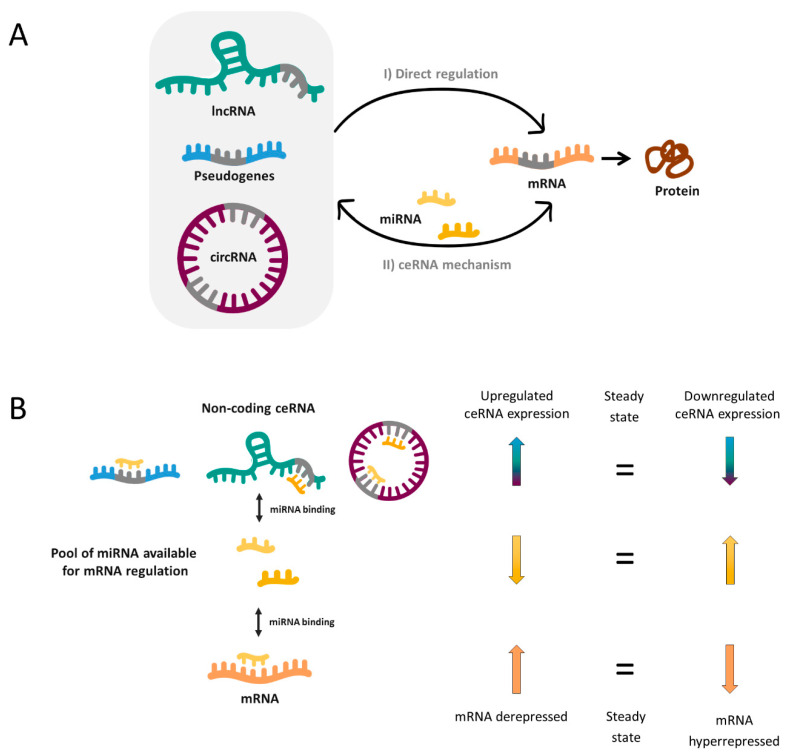
(**A**) Transcriptional and post-transcriptional regulation of messenger RNAs (mRNAs) (orange) can be both influenced by direct and indirect mechanisms involving long non-coding RNAs (lncRNAs) (green), pseudogenes (blue) and circular RNAs (circRNAs) (purple). (I) Direct mechanisms include some processes that act on the transcription rate in the nucleus through the specific RNA-RNA complex and others that help the stability of mRNA molecules in the cytoplasm. (II) Competing endogenous RNA (ceRNA) mechanism is a bidirectional indirect regulation mechanism mediated by microRNAs (miRNAs) (yellow). miRNAs bind lncRNAs, pseudogenes, circRNAs and mRNAs through the miRNA response elements (MRE) (grey). (**B**) ceRNA hypothesis. Upregulation of a certain ceRNA (pseudogene, lncRNA or circRNA) expression can decrease cellular concentrations of the corresponding miRNA, resulting in the de-repression of other transcripts (mRNA) that contains the same MREs (left arrows). Conversely, the downregulation of a certain ceRNA would lead to increased concentrations of specific miRNAs and thus to hyperrepression of mRNA expression (right arrows).

**Figure 2 ijms-21-09582-f002:**
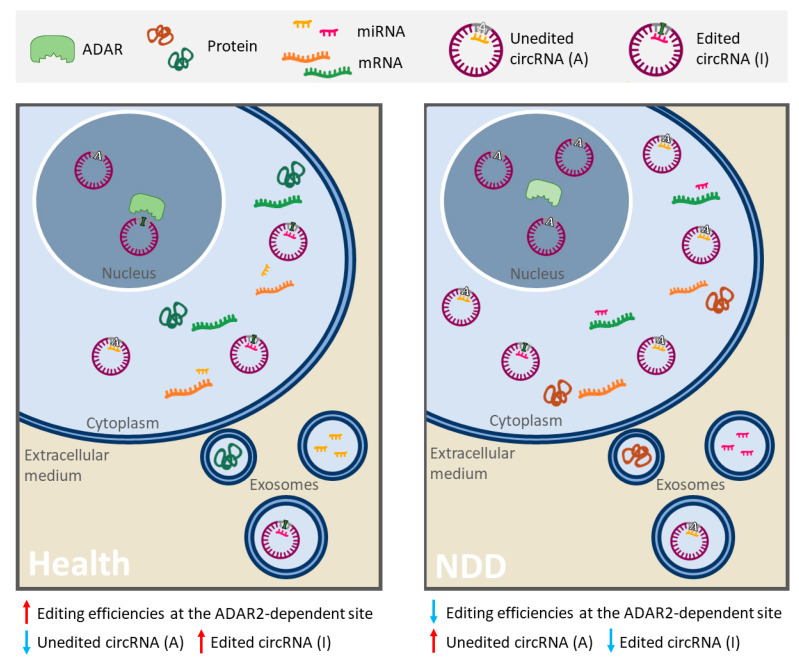
Schematic representation of alterations in RNA editing that could provide a specific RNA profile in neurodegenerative diseases (NDDs). In some linear and circular RNAs, the enzyme ADAR2 deaminates adenosine (A) into inosine (I), resulting in important biological consequences (especially in ncRNAs). On the one hand, a single editing site in MRE or miRNA seed region can drastically change its set of targets. In this image, circ-Purple acts as a miR-Yellow sponge, which regulates the mRNA expression of Orange gene (right panel). Deamination of A into I in circ-Purple could affect its binding site for miR-Yellow. In consequence, circ-Purple stops sponging miR-Yellow, and it may bind to another miRNA (miR-Pink) and promote the expression of Green gene (left panel). Hence, the ceRNA interaction network has changed, emerging a new or different regulatory axis. On the other hand, ADAR editing negatively regulates circRNA biogenesis, resulting in a decrease of circRNA levels (in the left panel there is less circ-Purple expression than in the right panel). In NDDs with a diminution of A-to-I RNA editing (like ALS, AD or PD), a different and opposite profile/pattern could be observed (right panel) with respect to a normal editing efficiency of ADAR (left panel). Therefore, alterations mediated by RNA editing in RNAs and its ceRNA interaction networks may serve as robust biomarkers of these NDDs. This figure is based on a previously published figure [292,296].

**Figure 3 ijms-21-09582-f003:**
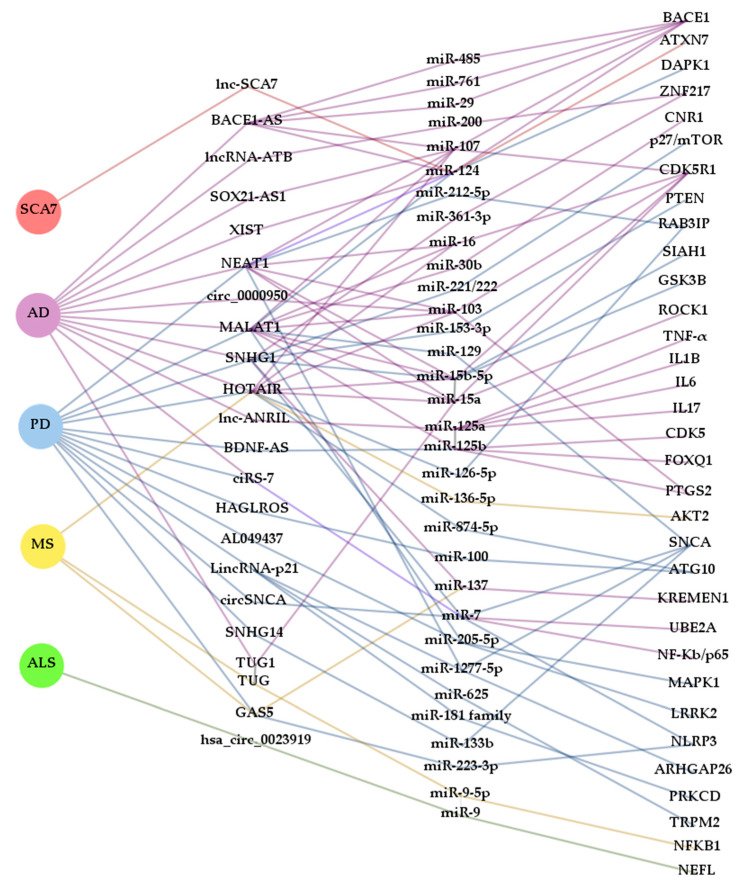
Complexity and interaction of ceRNETs in NDDs. The diagram was constructed with Gephi software from ceRNAs (lncRNAs and circRNAs) that, according to the bibliography cited in this review, contribute to the pathogenesis of more than one neurodegenerative disease and miRNAs that are part of ceRNETs from more than one ceRNA. Interactions between RNA molecules are represented with lines colored in accordance with the NDD background they have been described in: spinocerebellar ataxia type 7 (SCA7) (red), Alzheimer’s disease (AD) (purple), Parkinson’s disease (PD) (blue), multiple sclerosis (MS) (yellow) and amyotrophic lateral sclerosis (ALS) (green).

**Table 1 ijms-21-09582-t001:** miRNA-ceRNAs networks experimentally validated associated with NDDs.

Disease	ncRNA		miRNA	mRNA	Sample	Ref.
AD	lncRNA	BACE1-AS	miR-29, miR-485, miR-761, miR-124 and miR-107	*BACE1*	Computational analysis from human data and cellular and mouse models	[24]
			miR-214-3p	*-*		[25]
			miR-132-3p	*-*		[26]
		XIST	miR-124	*BACE1*	Cellular and mouse models	[27]
			miR-132	*-*		[28]
		NEAT1	miR-124	*BACE1*	Cellular and mouse models	[29]
			miR-107	*-*		[30]
		SOX21-AS1	miR-107	*-*	Cellular model	[31]
		NEAT1 HOTAIR MALAT1	miR-107, miR-103, miR-16, miR-195, miR-15a and miR-15b	*CDK5R1*	Cellular model	[32]
		MALAT1	miR-125b	*CDK5, FOXQ1* and *PTGS2*	Cellular and rat models	[33]
			miR-30b	*CNR1*		[34]
		TUG1	miR-15a	*ROCK1*	Cellular and mouse models	[35]
		SNHG1	miR-137	*KREMEN1*	Cellular model and human primary cell culture	[36]
			miR-361-3p	*ZNF217*		[37]
		lncRNA-ATB	miR-200	*ZNF217*	Cellular model	[38]
		LINC00094	miR-224-4p miR-497-5p	*SH3GL2*	Cellular model	[39]
		MIAT	miR-150-5p	*VEGF*	Cellular and mouse models	[40]
		Rpph1	miR-326	*PKM2*	Cellular and mouse models	[41]
			miR-122	*Wnt1*		[42]
			miR-330-5p	*CDC42*		[43]
		linc00507	miR-181c-5p	*MAPT* *TTBK1*	Cellular and mouse models	[44]
		lnc-ANRIL	mir-125a	*TNF-α*, *IL1B IL6* and *IL17*	Cellular model	[45]
	circRNA	ciRS-7	miR-7	*UBE2A*	Human brain	[46]
			* miR-7	* *NF-Κb/p65*	Cellular models	[47,48,49]
		circ_0000950	miR-103	*PTGS2*	Cellular models	[50]
		circHDAC9	miR-138	*Sirt1*	Cellular and mouse models	[51]
			miR-142-5p	*-*		[52]
PD	pseudogene	*GBAP1*	miR-22-3p	*GBA*	Cellular models	[53]
	lncRNA	SNHG1	miR-153-3p miR-15b-5p miR-7 miR-221/222	*PTEN* *SIAH1*, *GSK3β* *NLRP3* *CD*KN1B (p27)	Cellular and mouse models	[54] [55,56] [57] [58]
		HAGLROs	miR-100	*ATG10*	Cellular and mouse models	[59]
		HOTAIR	miR-874-5p	*ATG10*	Cellular and mouse models	[60]
			miR-126-5p	*RAB3IP*		[61]
		NEAT1	miR-212-5p	*RAB3IP*	Cellular models	[62]
			miR-1277-5p	*ARHGAP26*		[63]
			miR-124	*-*		[64]
		AL049437	miR-205-5p	*MAPK1*	Cellular and mouse models	[65]
		MALAT1	miR-205-5p	*LRRK2*	Cellular and mouse models	[66]
			miR-124	*DAPK1*		[67,68]
			miR-129	*SNCA* (α-syn)		[69]
		SNHG14	miR-133b	*SNCA*	Cellular and mouse models	[70]
		LincRNA-p21	miR-1277-5p	*SNCA*	Cellular and mouse models	[71]
			miR-181 family	*PRKCD* (PKC-δ)		[72]
			miR-625	*TRPM2*		[73]
		GAS5	miR-223-3p	*NLRP3*	Cellular and mouse models	[74]
		BDNF-AS	miR-125b-5p	*-*	Cellular and mouse models	[75]
		Mirt2	miR-101	*-*	Cellular model	[76]
		lncRNA H19	miR-301b-3p	*HPRT1*	Computational analysis from human data and cellular and mouse models	[77]
			miR-585-3p	*PIK3R3*		[78]
	circRNA	* ciRS-7	miR-7	*SNCA*	Cellular and mouse models	[79,80,81,82,83]
		circSNCA	miR-7	*SNCA*	Cellular model	[84]
		circzip-2	* miR-60	*M60.4ZK470.2*, *igeg-2* and *idhg-1*	Worm model	[85]
		circDLGAP4	miR-134-5p	*CREB*	Cellular and mouse models	[86]
MS	lncRNA	Gm15575	miR-686	*CCL7*	Cellular and mouse models	[87]
		PVT1	miR-21-5p	*SOCS5*	Cellular and mouse models	[88]
		TUG	miR-9-5p	*NFKB1 (p50)*	Cellular and mouse models	[89]
		HOTAIR	miR-136-5p	*AKT2*	Cellular and mouse models	[90]
		GAS5	miR-137	*-*	Human blood	[91]
	circRNA	hsa_circ_0106803	* miR-149	* *ASIC1a*	Human blood (PMBCs)	[92,93]
		hsa_circ_0005402 hsa_circ_0035560	* 14 miRNAs (miR-1248, miR-766)	*-*	Human blood (PMBCs)	[94]
SCA7	lncRNA	lnc-SCA7	miR-124	*ATXN7*	Human samples, and cellular and animal models	[95]

* Experimental validation is needed.

**Table 2 ijms-21-09582-t002:** miRNA-ceRNAs networks identified by transcriptome profiling studies and computational prediction associated with NDDs.

Disease	ceRNETs	Sample	Outcomes	Representative Networks	Ref.
AD	lncRNA-miRNA-mRNA	Mouse model (APP/PS1) brain (cortical samples) 12 months	One ceRNA network that included 4 lncRNAs, 5 miRNAs and 1082 mRNAs mainly related to AD-associated genes.	Rpph1/miR-326-3p, miR-330-5p/*Cdc42* C030034L19Rik/miR-182-5p/*Bdnf* Gm15477/miR-484/*Flnb* A830012C17Rik/miR-132-3p/*Smad4*	[43]
		Mouse model (APP/PS1) brain (cortical samples) 6 and 9 months	3 ceRNA networks built with lncRNAs, miRNAs and mRNAs differentially expressed according to the age at which they are found deregulated (6, 9 or 6 and 9 months).	LNC_000854/miR-122-5p/*Klf4* (6 months) LNC_000033/miR-128-2-5p, miR.135b-5p, miR.3097-3p, miR-31-5p, miR-449a-5p/*Synpo* (6 months) LNC_000217/miR-679-5p/*Akap5* (9 months) LNC_002639/miR-30B-5P/*Fyn* (6 and 9 months)	[96]
		Human brain (neurons from entorhinal cortex of mid-stage AD cases)	A neurofibrillary tangles-associated ceRNA network was built with 41 lncRNAs, 630 mRNAs and 2530 edges.	KB-1460A1.5/miR-302/*PTEN*	[97]
		Human brain (prefrontal cortex)	An AD-associated ceRNET containing 6 lncRNAs, 3 miRNAs and 91 mRNAs.	CERS6-AS1/miR-15B-5P/*PTEN* CTB-89H12.4/miR-155-5p/*CASP6*	[98]
	circRNA-miRNA-mRNA	Mouse model (SAMP8) brain 7 months	Two ceRNA networks built with circRNAs, miRNAs and mRNAs found differentially expressed.	6 circRNAs (mm10_circ_0027470, mm10_circ_ 0011311, mm10_circ_0018430, mm10_circ_0009478, mm10_circ_0010326, mmu_circ_0001442)/mmu-let-7g-3p/*Hmgb2* 5 circRNAs (mmu_circ_0000967, mmu_ circ_0001293, mm10_circ_0027491, mm10_ circ_0027459, mm10_circ_0027483)/miR-122-5p/*Dio2*	[99]
		Mouse model (Tg2576) brain 7 and 12 months	Four ceRNA networks built with circRNAs, miRNAs and mRNAs found differentially expressed.	mmu_circ_37345/miR_335-3p/*SLY* (12 months)	[100]
		Rat model (Aβ1-42) brain (hippocampal samples)	A ceRNA network built with 140 circRNAs, 140 miRNAs and 20 mRNAs with 503 relationships.	circ_101834 and circ_004690/miR-7a-5p/*Aqp3*	[101]
AD	circRNA-miRNA-mRNA	Mouse model (SAMP8) brain (hippocampal samples) 5 and 10 months	A ceRNA network predicted.	mmu_circ_017963/5 miRNAs (mmu_miR_1896, mmu_miR_1955-5p, mmu_miR_7030-3p, mmu_ miR_7033-3p, mmu_miR_542-5p)/313 mRNAs	[102]
		Mouse model (SAMP8) brain (hippocampal) 5 months	Two ceRNETs built with 2 circRNAs found differentially expressed in PNS-treated mice: mmu_circ_013636, 5 miRNAs and 442 mRNAs, mmu_circ_012180, 5 miRNAs and 631 mRNAs.	mmu_circ_012180/miR_6972-5p/*Gsdmd*	[103]
		Mouse model (APP/PS1) brain (cortex) 6 and 9 months	Five ceRNA networks were constructed based on differentially expressed circRNAs, miRNAs and mRNAs.	6 circRNAs (mmu_circ_0000452, mmu_circ_0000453, novel_circ_0010838, novel_circ_0011428, novel_circ_ 0033961, novel_circ_0037760)/mmu-miR-466b-5p/*Scube2* 4 miRNAs (mmu-miR-219b-5p, mmu-miR-350-5p, mmu-miR-450b-5p, mmu-miR-9b-5p)/*Sorbs2* 7 circRNAs (mmu_circ_0000433, novel_circ_ 0019965, mmu_circ_0001473, novel_circ_0021924, novel_circ_0028455, novel_circ_0051361, novel_ circ_0058143)/mmu-miR-122b-3p/*Cntnap2* Mmu_circ_0044900/4 miRNAs (mmu-miR-449a-5p, mmu-miR-467a-3p, mmu-miR-540- 3p, and mmu-miR-669f-3p)/*Creb*	[104]
		Mouse model (5xFAD) pineal gland 5 months	A circRNA-miRNA network was constructed with 10 circRNAs. From it, a complete ceRNA net was predicted.	circMboat2 and circNlrp5-ps/miR-483/*Aanat*	[105]
		Human brain	An AD-associated ceRNET was constructed with 276 circRNAs, 14 miRNAs and 1117 mRNAs. AD risk ceRNET of KIAA1586 was stablished with 3 miRNAs (hsa-miR-29b, hsa-miR-101 and hsa-miR-15a) and 159 mRNAs.	KIAA1586/miR-15a/*PSEN2* KIAA1586/miR-101/*UBE2A* KIAA1586/miR-15a, miR-29b/*BACE1*	[106]
		Human brain	A circRNA-mRNA co-expression network was predicted.	circHOMER1/miR-651/*PSEN1* and *PSEN2* circCORO1C/miR-105/*APP* and *SNCA*	[107]
AD	circRNA-miRNA-mRNA	Human brain	A circRNA-mRNA co-expression network was predicted	hsa_circ_0000994 and hsa_circ_0005232 (from *SLC8A1* gene)/-/*EPHA4*	[108]
		Human cerebrospinal fluid	A circRNA-miRNA network was built with the top 5 up- and down-regulated circRNAs (circ-LPAR1, circ-AXL, circ-GPHN, circ-ITPR3, circ-GPI, circ-HAUS4, circ-KIF18B, circ-ATP9A, circ-PCCA and circ-TTC39C).	circ-TTC39C/miR-210-3p/*BDNF* circ-PCCA/miR-138-5p/*RARA*	[109]
		Human blood (PBMCs)	Four nets constructed: (I) circRNA-miRNA network with the top 10 up- and down-regulated circRNAs (II) ceRNA network of hsa_circ_082547 (III) ceRNA network with 3 circRNAs (hsa_circ_101618, hsa_circ_405619, and hsa_circ_000843), 15 miRNAs and 223 mRNAs. (IV) ceRNA network of 4 circRNAs (hsa_circ_402265, hsa_circ_061346, hsa_circ_405836 and hsa_circ_061343), 20 miRNAs and 576 mRNAs.	hsa_circ_061346/hsa-miR-5916-3p/*APP* hsa_circ_000843/hsa-miR-335-3p/*SLC8A1* hsa_circ_061346/hsa-miR-103a-2-5p/*HOMER1* 13 circRNAs (hsa_circ_101367, hsa_circ_101368, hsa_circ_103729, hsa_circ_406440, hsa_circ_ 101726, hsa_circ_103394, hsa_circ_100861, hsa_ circ_102448, hsa_circ_103548, hsa_circ_037274, hsa_circ_101159, hsa_circ_104137, hsa_circ_101740)/miR-455-3p/*APP*	[110]
PD	lncRNA-miRNA-mRNA	PD cell model (SY-SH5Y cells treated with α-synuclein oligomer)	PD-associated ceRNET that included the lncRNAs AC009365.4, RPS14P3 and G046036 together with the mRNAs *IRF1, RIMKA, NAV1, SACS* and *SDC2*.		[111]
		Human blood	A ceRNA regulatory network, including 7 lncRNAs (XIST1, PART1, MCF2L2, NOP14-AS1, LINC00328, LINC00302 and FAM215A), 3 miRNAs (miR-7, miR-433 and miR-133b), and 55 mRNAs.	XIST1/miR-7/IGF1R, RKAG2, RAD51 and ITCH XIST1/miR-433/CA12, CTH and PRKACA PART1/miR-133b/IGF1R and CTH	[112]
		Human brain (substantia nigra)	A ceRNA network associated with PD, that included 9 lncRNAs, 18 miRNAs, and 185 mRNA.	XIST/miR-615-3p/mRNAs NEAT1/miR-124-3p/mRNAs	[113]
	circRNA-miRNA-mRNA	Mouse model (MPTP) brain	A ceRNET built with 6 circRNAs (circ_0003292, circ_0001320, circ_0005976, circ_0005388, circ_0012384, and circ_003328), 13 miRNAs and 112 mRNAs.	mmu_circ_0003292/miR-132/*Nr4a2* mmu_circ_0001320/miR-124/*Sox9*	[114]
		Human brain	A circRNA-associated ceRNA network predicted and validated.	circSLC8A1/miR-128/*BMI1*, *SIRT1* and *AXIN1*	[115]
ALS	circRNA-miRNA-mRNA	Human blood (leukocytes)	Two networks predicted.	hsa_circ_0023919/miR-9/*NEFL* hsa_circ_0063411/miR-647/-	[116,117]

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
