# Peer review of "Competing Endogenous RNA Networks as Biomarkers in Neurodegenerative Diseases"

_ijms, 2020, doi:10.3390/ijms21249582_

Round 1

Reviewer 1 Report

In the paper of  Leticia Moreno-García et al., the author present a comprehensive review on

 Competing endogenous RNA networks in neurodegenerative diseases and discuss these networks in context of possible biomarkers for diagnosis, disease monitoring, elucidating of molecular mechanism and distinguishing between different diseases. In the paper important data are collected and the paper thus represent an interesting and useful contribution.

However, several points need correction and improvements:   

Figure 1 A needs more explanation. Please explain gray sections in different RNAs. The arrow in Fig.1A, II) (ceRNA mechanism) should also point in direction to noncoding RNAs.

Figure 2 for miRNA presentation more contrast colors should be selected

There is no data how Figure 3 was constructed. Please explain an add.

Throughout the whole paper there are very many spelling and grammatical errors and a lot of sentences are incomprehensible and should be improved and corrected, for example like the one:

“In fact, it has been reported that some ncRNAs, like ciRS-7 or MALAT1, are also detected in exosomes, in which their levels can change in function of cellular miRNA abundance and they may their biological activity to regulate miRNA expression in target cells after exosomal delivery [269,270]”.

Above all, there are a lot of inconsistencies in literature citations. For example, for the whole following chapter the references are completely wrong:

“V Similar to lncRNAs, little is known about circRNA in ALS. However, three circRNAs have shown some potential as diagnostic biomarkers [257]. Dolinar et al. [257] investigated expression profile of circRNAs and identified 425 differentially expressed in leukocytes from sporadic ALS (SALS) patients. Among these, hsa_circ_0023919, hsa_circ_0063411 and hsa_circ_0088036 showed the highest significance as well as clinical relevance. Curiously, the expression levels of several circRNAs were positively correlated with each other, suggesting that they could be involved in similar biological processes and/or co-regulated [257]. Although possible circRNA interaction networks were not investigated, the authors predicted miRNA targets for two affected circRNAs. Hsa_circ_0023919 was found significantly down-regulated in SALS patients and contains two binding sites for hsa-miR-9. Accordingly, two previous studies confirmed the upregulation of miR-9 in both mouse model of ALS [258] and in blood samples of ALS patients [259]. It has been reported that miR-9 directly binds to and reduces the mRNA levels of NEFL [260], which encodes the neurofilament light polypeptide. Interestingly, aggregation of intermediate filament is a characteristic ALS hallmark. In light of this evidence, one potential biomarker based on ceRNA axis has been proposed in ALS (hsa_circ_0023919/miR-9/NEFL), where hsa_circ_0023919 could act as miR-9 sponge and regulate metabolism of intermediate filaments (NEFL) observed in ALS [261]. In contrast, hsa_circ_0063411 was found upregulated in patients. This circRNA contains one binding site for miR-647, which has been seen expressed in spinal cords from healthy subjects, but not from sALS patients [262]. However, the study did not establish a mRNA target of hsa_circ_0063411/ miR-647 axis, so further studies are necessary to elucidate this potential ALS ceRNA network. One possibility is that hsa_circ_0063411 may bind to miR-647 and increase the expression of PTEN, a direct target of miR-647 [263]. As mentioned above, PTEN has been seen involved in AD- and PD-associated ceRNA networks. Since PTEN has been reported as a potential therapeutic target in motor neuron diseases, including ALS or SMA (spinal muscular atrophy) [264–266], it remains possible that PTEN mRNA could also be part of ALS-linked ceRNETs”.

  1. Daniel, C.; Lagergren, J.; Öhman, M. RNA editing of non-coding RNA and its role in gene regulation. Biochimie. 2015, 117, 22-27.
  2. Shevchenko, G.; Morris, K V. All I’s on the RADAR: role of ADAR in gene regulation. FEBS Lett. 2018, 592(17), 2860-2873. doi:10.1002/1873-3468.13093
  3. Nigita, G.; Distefano, R.; Veneziano, D.;et al. Tissue and exosomal miRNA editing in Non-Small Cell Lung Cancer. Sci Rep. 2018, 8(1), 10222. doi:10.1038/s41598-018-28528-1
  4. Hideyama, T.; Yamashita, T.; Aizawa, H.;et al. Profound downregulation of the RNA editing enzyme ADAR2 in ALS spinal motor neurons. Neurobiol Dis. 2012, 45(3), 1121-1128. doi:10.1016/j.nbd.2011.12.033
  5. Aizawa, H.; Hideyama, T.; Yamashita, T.;et al. Deficient RNA-editing enzyme ADAR2 in an amyotrophic lateral sclerosis patient with a FUS(P525L) mutation. J Clin Neurosci. 2016, 32, 128-129. doi:10.1016/j.jocn.2015.12.039
  6. Hosaka, T.; Yamashita, T.; Teramoto, S.; Hirose, N.; Tamaoka, A.; Kwak, S. ADAR2-dependent A-to-I RNA editing in the extracellular linear and circular RNAs. Neurosci Res. 2019, 147, 48-57. doi:10.1016/j.neures.2018.11.005
  7. Khermesh, K.; D’Erchia, AM.; Barak, M.;et al. Reduced levels of protein recoding by A-to-I RNA editing in Alzheimer’s disease. RNA. 2016, 22(2), 290-302. doi:10.1261/rna.054627.115.
  8. Singh, M. Dysregulated A to I RNA editing and non-coding RNAs in neurodegeneration. Front Genet. 2013, 3, 326. doi:10.3389/fgene.2012.00326

Reviewer 2 Report

These is a nice review showing key examples and points of competing endogenous RNA networks as biomarkers in different neurodegenerative diseases.

I just have a minor comment. In the following statement:

“Without doubt, the reality is more complex and a miRNA can bind more than one mRNA (50% of miRNAs are predicted to target 1-400 mRNAs and some of them up to 1.000)” . Could the authors add a reference to this statement? It seems arbitrary otherwise.

Round 2

Reviewer 1 Report

Dear authors,

There are still some minor spelling errors, please correct.

Please also add at the beginning of Materials and Methods section from which data bases and the time interval for literature selection for this review article.
